

# Environmental dissemination of respiratory viruses: dynamic interdependencies of respiratory droplets, aerosols, aerial particulates, environmental surfaces, and contribution of viral re-aerosolization

M. Khalid Ijaz[1], Syed A. Sattar[2], Raymond W. Nims[3], Stephanie A. Boone[4], Julie McKinney[1] and Charles P. Gerba[4]

[1] Global Research & Development for Lysol and Dettol, Reckitt Benckiser LLC, Montvale, NJ, United States of America
[2] Department of Biochemistry, Microbiology & Immunology, Faculty of Medicine, University of Ottawa, Ottawa, Ontario, Canada
[3] Syner-G BioPharma, Boulder, CO, United States of America
[4] Water & Energy Sustainable Technology Center, University of Arizona, Tucson, AZ, United States of America

Corresponding author
M. Khalid Ijaz,
khalid.ijaz@reckitt.com

## ABSTRACT

During the recent pandemic of COVID-19 (SARS-CoV-2), influential public health agencies such as the World Health Organization (WHO) and the U.S. Centers for Disease Control and Prevention (CDC) have favored the view that SARS CoV-2 spreads predominantly *via* droplets. Many experts in aerobiology have openly opposed that stance, forcing a vigorous debate on the topic. In this review, we discuss the various proposed modes of viral transmission, stressing the interdependencies between droplet, aerosol, and fomite spread. Relative humidity and temperature prevailing determine the rates at which respiratory aerosols and droplets emitted from an expiratory event (sneezing, coughing, *etc.*) evaporate to form smaller droplets or aerosols, or experience hygroscopic growth. Gravitational settling of droplets may result in contamination of environmental surfaces (fomites). Depending upon human, animal and mechanical activities in the occupied space indoors, viruses deposited on environmental surfaces may be re-aerosolized (re-suspended) to contribute to aerosols, and can be conveyed on aerial particulate matter such as dust and allergens. The transmission of respiratory viruses may then best be viewed as resulting from dynamic virus spread from infected individuals to susceptible individuals by various physical states of active respiratory emissions, instead of the current paradigm that emphasizes separate dissemination by respiratory droplets, aerosols or by contaminated fomites. To achieve the optimum outcome in terms of risk mitigation and infection prevention and control (IPAC) during seasonal infection peaks, outbreaks, and pandemics, this holistic view emphasizes the importance of dealing with all interdependent transmission modalities, rather than focusing on one modality.

## INTRODUCTION

The COVID-19 (SARS-CoV-2) pandemic has resulted in a re-examination of pathways for respiratory virus dissemination and transmission. The transmission of respiratory viruses, such as severe acute respiratory coronavirus 2 (SARS-CoV-2) or influenza virus, from an infected to an uninfected person has been characterized as occurring mainly through "close contact" with an infected person (*United States Centers for Disease Control and Prevention, 2019*). This rather vague statement can be interpreted to include transmission by "close contact" or actual direct physical contact between infected and susceptible individuals, and indirect contact through an intermediate surface (*Li, 2020*). Direct physical contact is not in scope for this article. The consensus view is that exposure to respiratory droplets from the infected person, and not physical contact, (*e.g.*, shaking hands) is the primary transmission mode for respiratory viruses, such as SARS-CoV-2 (*United States Centers for Disease Control and Prevention, 2019*; *World Health Organization, 2020*). Exposure to respiratory droplets might appear to be a straightforward mode of pathogen transmission; however this is highly complicated when the continuum of infectious droplet size and the physical changes in expired droplets (*e.g.*, evaporation) and matrices (mucus, saliva, dust, *etc.*) to which the infectious droplets might be adsorbed are factored in. Viruses are expelled *via* respiratory droplets, which are typically categorized according to size. Droplet size (diameter) plays an important role in infectious droplet dissemination, deposition onto fomites in the environment, and access to and therefore viral infectivity in the human respiratory system (*Drossinos, Weber & Stilianakis, 2021*). Direct viral transmission by droplets has been characterized as short-time and short-range in nature. The duration of time and the range over which the transmission may take place is impacted by several factors, including the size of the droplets, environmental factors (relative humidity, temperature), gravitational settling rate, and dispersion in turbulent air jets (*Ijaz et al., 1985*; *Drossinos, Weber & Stilianakis, 2021*; *Foat et al., 2022*). Droplet size should be viewed as a spectrum, and intermediate sized infectious droplets may behave more like infectious aerosols than large droplets in this regard (*Norvihoho et al., 2023*). This fact is key to arriving at conclusions regarding the relative importance of infectious droplets *vs.* infectious aerosols in transmitting viral infections.

Long-range (distal) dissemination of viruses *via* respiratory aerosols (size: <5 µm) also has been advocated as an important transmission mechanism (*Morawska & Cao, 2020*; *Zhang & Duchaine, 2021*; *Leung, 2021*; *Meyerowitz et al., 2021*; *Samet et al., 2021*). Indirect transmission through the intermediacy of hands and contaminated environmental surfaces has not been considered as important a mechanism for dissemination and transmission of respiratory viruses due to lack of clinical evidence. However, not all experts in the field are in agreement in this regard (*Ijaz, Nims & McKinney, 2021a*; *Leung, 2021*). The possibility of re-aerosolization or resuspension of virus from contaminated environmental

surfaces deserves attention, as a contributor to airborne virus content and subsequent dissemination of such viruses (*Stephens et al., 2019*; *Leung, 2021*). Finally, the potential for long-range transmission of respiratory viruses on aerial particulate matter (such as dust, allergens, $PM_{10}$, $PM_{2.5}$, $PM_1$, *etc.*) has been proposed (*Asadi et al., 2020*; *Barakat, Muylens & Su, 2020*; *Bashir et al., 2020*; *Farhangrazi et al., 2020*; *Setti et al., 2020*; *Tung et al., 2021*; *Betsch & Sprengholz, 2021*; *Brandt & Mersha, 2021*; *Cao et al., 2021a*; *Chen, 2021*; *Leung, 2021*; *Santurtún et al., 2022*; *Solimini et al., 2021*; *Tchicaya et al., 2021*; *Yilbas et al., 2021*; *Baron, 2022*). The latter mechanism of dissemination of viruses is of interest, due to the ability of certain of the aerial particulate types to upregulate cellular receptors involved in viral infection (*e.g.*, the human ACE2 receptor in the case of SARS-CoV-2), as will be discussed later in this review.

Our goal in this review is to take a neutral position in this debate and to define the potential dissemination/transmission pathways and the major arguments for and against the relevance of these pathways. Part of the confusion and the difficulty in resolving the ongoing debate on transmission mechanisms is semantic (*Shiu, Leung & Cowling, 2019*; *Li, 2020*). That is, the term "airborne" has commonly been used to denote infectious aerosol dissemination, and not dissemination through respiratory droplets, even though both involve dispersion of virus in the air. Secondly, there are numerous interdependencies in the various viral transmission modalities (Fig. 1). Namely, respiratory droplets may be transmitted directly or may fall and deposit on environmental surfaces and, therefore, contribute to the indirect transmission pathway. Viruses on fomite surfaces can be resuspended in the air through various human activities (*e.g.*, walking, opening doors, vacuuming, *etc.*), contributing to the infectious droplet/aerosol transmission pathway. Aerosol transmission involves virus from evaporated respiratory droplets that form droplet nuclei, aerosols generated through talking, sneezing, coughing, *etc.*, virus re-aerosolized from surfaces, and virus conveyed on aerial particulate matter (dust, allergens, PM, *etc.*).

In this review, we have categorized dissemination pathways according to distance and temporal (time) risk to help clarify the semantic problems associated with various transmission modalities for respiratory viruses (Fig. 2). Clear definitions of what is meant by "short range", "long range", "short time", "long time", *etc.* are not to be found in the literature. Typically, however, the term "short range" denotes 6 ft or less than 1.5 m (*Li, 2020*). Short time is the time it typically takes for respiratory droplets to settle under gravity (seconds to minutes, depending upon size) (*Knight, 1990*; *Castaño et al., 2021*). The upper limits of distance and time over which viral aerosols might remain infectious have yet to be empirically determined and likely depend on the virus itself. But long-range transmission has been characterized as greater that >1.5 m (*Li, 2020*). We and others (*Killingley & Nguyen-Van-Tran, 2013*; *Leung, 2021*; *Li, 2020*) make the argument that each of the primary transmission pathways (respiratory droplets, viral aerosols, and indirect transmission *via* contaminated fomites; Fig. 1) are relevant, and display interdependencies. Because of this, it seems inappropriate to ignore any one of these pathways from the standpoint of assessing and mitigating risk of acquiring a respiratory viral infection.
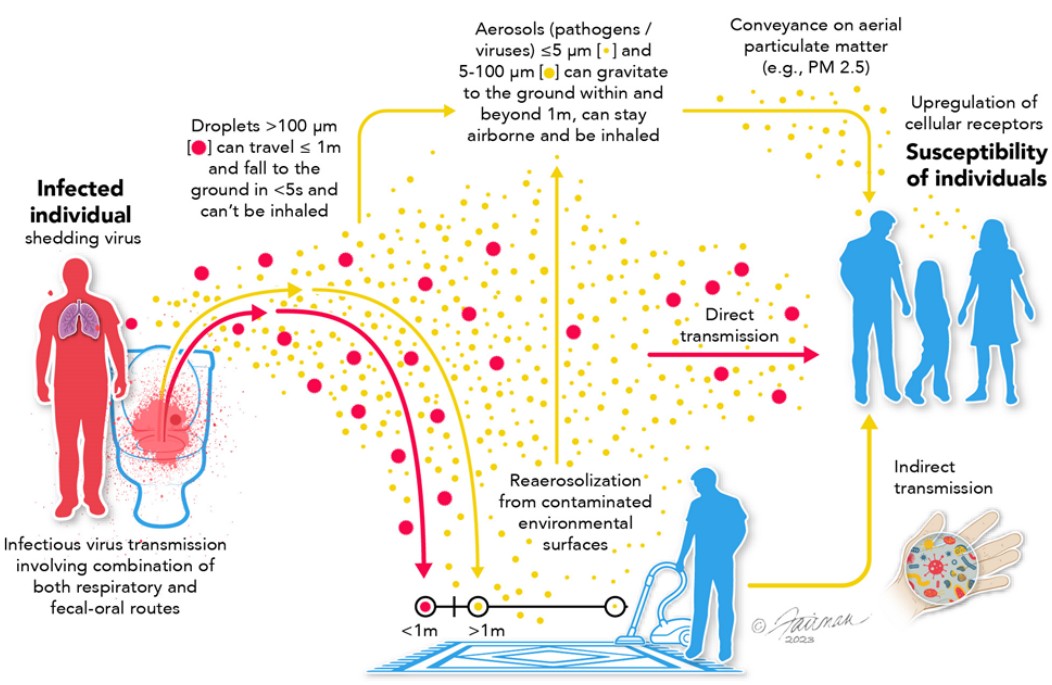

**Figure 1** Possible routes of transmission of respiratory viruses, such as influenza, MERS-CoV, and SARS-CoV-2, showing interdependencies. Modified from *Ijaz et al., 2020*. Data credit: *Wang et al., 2021*; *Woodby, Arnold & Valacchi, 2021*; and *Gu et al., 2023*. CC BY 4.0. https://creativecommons.org/licenses/by/4.0.

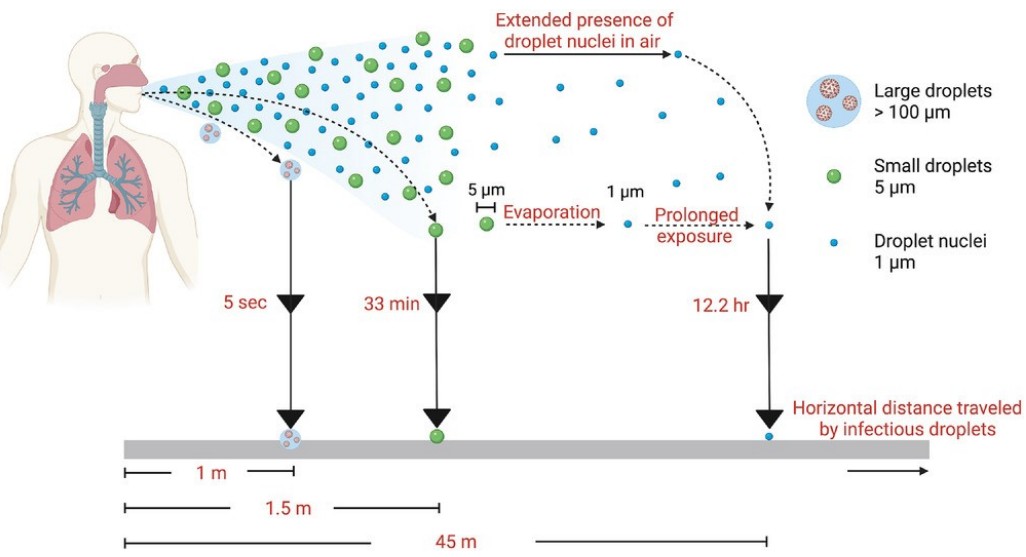

**Figure 2** Range and time risk associated with different virus transmission pathways. Image source: *Joseph et al. (2022)*, CC BY 4.0, https://creativecommons.org/licenses/by/4.0.

# SURVEY METHODOLOGY

This survey is intended to provide a sampling of the literature emphasizing the different pathways of transmission of respiratory viruses, including the relative importance of each proposed mode, and the interdependencies between the different modes. As such, our methods for ascertaining relevant literature were not intended to access all of the relevant literature, and exclusion criteria were not applied when accessing the literature. Instead, we started with the World Health Organization's position on transmission modes, articulated in June 2020 (*World Health Organization, 2020*), then searched the relevant literature in Google Scholar and PubMed on the topics covered in the review (see abstract and the subject headings below). Search topics used included SARS-CoV-2, other coronaviruses and respiratory viruses; droplet transmission, airborne or aerosol transmission, fomite or indirect transmission, re-aerosolization of virus, viral conveyance ("hitchhiking") on aerial particulate matter, and methods for mitigating transmission of airborne virus. As much of the recent literature on these subjects was motivated by the recent SARS-CoV-2 pandemic, the cited literature is weighted toward SARS-CoV-2 as the challenge virus, and accordingly, tended to be very recent (*i.e.,* since late 2019). The target audience for this review includes the infection prevention and control community, regional public health authorities worldwide, and the aerobiology research and epidemiology communities.

## Short-range and -time: viral dissemination by respiratory droplets

The WHO update (*World Health Organization, 2020*) on SARS-CoV-2 transmission modes states "Transmission of SARS-CoV-2 can occur through direct, indirect, or close contact with infected people through infected secretions such as saliva and respiratory secretions or their respiratory droplets, which are expelled when an infected person coughs, sneezes, talks or sings. Respiratory droplets are >5–10 μm in diameter whereas droplets <5 μm in diameter are referred to as droplet nuclei or aerosols. Respiratory droplet transmission can occur when a person is in close contact (within 1 m) with an infected person who has respiratory symptoms (*e.g.,* coughing or sneezing) or who is talking or singing; in these circumstances, respiratory droplets that include virus can reach the mouth, nose or eyes of a susceptible person and can result in infection." In the categorization of *Li (2020)* and *Marr & Tang (2021)*, this transmission mode corresponds to "drop spray" or "spray" transmission" and is considered short range and short time. On the basis of knowledge gained with other respiratory viruses, such as SARS-CoV and influenza viruses, direct exposure to infectious respiratory droplets is considered the primary transmission mechanism for SARS-CoV-2 and respiratory viruses in general (*Leung, 2021*). The acquisition of viral infections through exposure to infectious respiratory droplets (*i.e.,* size > 100 μm) (*Leung, 2021*; *Marr & Tang, 2021*) is likely, since inhalation of such infectious droplets released from an infected individual may result in direct exposure of susceptible oral, nasal, and ocular mucous membranes containing the angiotensin-converting enzyme 2 (ACE2) receptor (in case of SARS-CoV-2) in a nearby susceptible individual (*Hoffmann et al., 2020*; *Salamanna et al., 2020*; *Yan et al., 2020*).

Virus content and survival have been shown to be related to droplet particle size, with survival of transmissible gastroenteritis virus (*Coronaviridae*) and swine influenza virus

and avian influenza virus (*Orthomyxoviridae*) being enhanced at larger particle sizes (*e.g.,* 300–450 nm), relative to survival at particle sizes closer to that of the virions themselves (*i.e.,* 100–200 nm) (*Zuo et al., 2013*). It should be clarified that the virus in such droplets is not typically present in the form of free virions, but is typically solvated, likely within a physiological matrix consisting of respiratory fluid, mucus, or saliva, which might further enhance survival (*Kampf et al., 2020b*; *Rasheed et al., 2021*; (*Zhang & Duchaine, 2021*)). The size distribution of respiratory droplets has been evaluated (*e.g.,* *Morawska, 2006*). Infectious respiratory droplets are heavy enough to settle under gravity, within a relatively short distance from the infected person (*Leung, 2021*; *Wang et al., 2021*), hence the terms "close, proximal, or short-range" for this mechanism of viral dissemination (Fig. 2).

Respiratory droplets emitted from infected individuals can be viewed as one of the primary sources for virus disseminated through other modes of viral transmission. For instance, the evaporation of infectious respiratory droplets to form droplet nuclei contributes to the generation of viral aerosols (*Morawska & Cao, 2020*; *Zhang & Duchaine, 2021*). As mentioned previously, intermediate sized droplets may also contribute to or behave like aerosols (*Norvihoho et al., 2023*). In addition, the gravity-induced deposition of respiratory droplets onto fomites contributes to the indirect transmission pathway (*Zhang et al., 2020*; *Zhang & Duchaine, 2021*). Larger respiratory droplets containing virus may not be able to reach the lungs during inhalation (*Lamers & Haagmans, 2022*), but certainly can access the susceptible membranes of the nasal, ocular, and oral mucosa (*Zhang et al., 2020*; *Zuo, Uspal & Wei, 2020*). This distinction is likely not applicable to smaller or intermediate sized droplets. This appears to be sufficient for initiation of SARS-CoV-2 infections, and for other respiratory viral infections, as the expression of ACE-2 receptors appears to be highest in the upper respiratory tract (*Hou et al., 2020*). But is this the predominant route, and can this route be considered the only important route for dissemination of respiratory viruses? If we mitigate short-range respiratory droplet transmission, will this be sufficient to prevent virus dissemination to susceptible individuals during outbreaks and pandemics?

## Short- to long-range and time dissemination through respiratory aerosols

The relative importance, to transmission of respiratory viruses, of short- or long-range (distal) transmission *via* respiratory aerosols has also been the subject of scientific debate, which in fact predates the SARS-CoV-2 pandemic (*Sattar & Ijaz, 1987*; *Shiu, Leung & Cowling, 2019*; *Brosseau, 2020*; *Cimolai, 2020*; *Morawska & Cao, 2020*; *Zhang & Duchaine, 2021*; *Cao et al., 2021a*; *Chen, Jia & Han, 2021*; *Dinoi et al., 2022*; *Leung, 2021*; *Meyerowitz et al., 2021*; *Samet et al., 2021*). The WHO stance on aerosol transmission of SARS-CoV-2, for instance, is as follows: "WHO, together with the scientific community, has been actively discussing and evaluating whether SARS-CoV-2 may also spread through aerosols in the absence of aerosol generating procedures, particularly in indoor settings with poor ventilation. The physics of exhaled air and flow physics have generated hypotheses about possible mechanisms of SARS-CoV-2 transmission through aerosols. These theories suggest that (1) a number of respiratory droplets generate microscopic aerosols (<5 µm) by evaporating, and (2) normal breathing and talking results in exhaled aerosols. Thus, a

susceptible person could inhale aerosols, and could become infected if the aerosols contain the virus in sufficient quantity to cause infection within the recipient. However, the proportion of exhaled droplet nuclei or of respiratory droplets that evaporate to generate aerosols, and the infectious dose of viable SARS-CoV-2 required to cause infection in another person are not known, but it has been studied for other respiratory viruses." (*World Health Organization, 2020*).

The latter are curious speculations, as these statements imply that viral aerosols are only generated through evaporation of respiratory droplets, which seems incorrect. Theoretically, infectious viral aerosols may be generated directly from infected individuals by speaking, coughing, sneezing, *etc.*, or from other activities, including but certainly not limited to, flushing or cleaning toilets (*Li, Wang & Chen, 2020*; *Meng et al., 2020*; *Vardoulakis, Espinoza Oyarce & Donner, 2022*; *Goforth et al., 2023*), and may be generated through re-aerosolization from contaminated fomites, as will be discussed later in this review (*Khare & Marr, 2015*; C. Gerba, 2023, unpublished). The extent to which this is true, and the relative quantity of infectious virus that might be conveyed on such aerosols, need further confirmation in empirical studies. Recent papers have suggested that relatively low numbers of infectious viruses (*e.g.*, as low as 10 $TCID_{50}$ of a wild-type virus (SARS-CoV-2/human/GBR/484861/2020)) are required, as reported for volunteers without previous SARS-CoV-2 infection or vaccination (*Killingley et al., 2022*). The relatively low infectious dose for SARS-CoV-2 has been observed also in clinical and modeling studies (*Prentiss, Chu & Berggren, 2022*; *Stettler et al., 2022*). These data suggest that suspended aerosols contain sufficient amounts of virus to initate infection.

In the categorization of *Li (2020)* and *Marr & Tang (2021)*, aerosol inhalation transmission denotes infectious virus-laden respiratory droplets that remain suspended in the air. This is considered a short-range and short-time transmission pathway (*via* proximity inhalation), although long-range, long-time transmission is also possible (*via* distant inhalation) (*Li, 2020*) (Fig. 2). Arguments in favor of aerosol transmission include the demonstrated persistence of infectious respiratory virus in aerosols (*Ijaz et al., 1985*; *Sattar & Ijaz, 1987*; *Zuo et al., 2013*; *Kormuth et al., 2018*; *van Doremalen et al., 2020*; *Fears et al., 2020*; *Schuit et al., 2021*; *Zarger et al., 2021*) and viral genomic material or infectious virus detection in air samples (*Chia et al., 2020*; *Liu et al., 2020*; *Borges et al., 2021*; *Dumont-Leblond et al., 2021*; *Kotwa et al., 2021*; *Lednicky et al., 2021*; *Dinoi et al., 2022*). Viral persistence in aerosols has been found to depend on the pH of the carrier infected patient's body fluid (*Joonaki et al., 2020*; *Luo et al., 2023*), as well as the temperature and the relative humidity (RH) of the air indoor (*Ijaz et al., 1985*; *Karim et al., 1985*; *Sattar & Ijaz, 1987*; *Smither et al., 2020*). In addition, the particle sizes characteristic of infectious aerosols enable these to impact not only susceptible oral, nasal, and ocular mucous membranes, but also the lower respiratory tract and lung tissue (*Morawska et al., 2009*; *Zuo, Uspal & Wei, 2020*). Studies have been performed to simulate the spread of viral aerosols through model indoor spaces (hospital closed transfusion room; *Cao et al., 2021b*; classroom; *Rencken et al., 2021*). Additional modeling has been performed to help understand deposition of aerosolized virus onto fomites (*Zuo, Uspal & Wei, 2020*; *Wang et al., 2022*).

Several respiratory viruses have been proposed to be transmitted through aerosols, including SARS-CoV, Middle East respiratory syndrome virus, and influenza virus (*Tang et al., 2020* and references therein). Additional arguments in favor of long-range transmission *via* aerosols include the fact that aerosols may remain dispersed and airborne over longer distances and for longer durations than do droplets, albeit with expected dilution with increased distance from the source (*Castaño et al., 2021*). Aerosol transmission also might be expected to result in accumulation, over time, of viruses within confined indoor spaces, provided that the aerosolized virus remains airborne, and does not settle. Transmission by aerosols should be favored under conditions of relatively crowded and confined spaces with poor ventilation, where naturally aerosolized infectious virus emitted from infected individuals (pre-symptomatic, symptomatic, post-symptomatic including asymptomatic) might be expected to be maximized (*Tang et al., 2020*). However, these are also the same conditions that favor direct transmission by intermediate sized droplets and indirect transmission by fomites (*Otter et al., 2016*; *Jones, 2020*; *Kraay et al., 2021*). SARS-CoV-2 infectivity in such aerosols has been demonstrated to depend on the aerosol matrix and the RH and temperature (*Smither et al., 2020*), as has been observed previously with other enteric and respiratory viruses, including coronaviruses (*Ijaz et al., 1985*; *Karim et al., 1985*; *Sattar & Ijaz, 1987*). The infectivity half-life of SARS-CoV-2 in a culture medium or saliva aerosol matrix has been reported to be 75 and 42 min, respectively, at medium RH, and 30 and 177 min, respectively, at high RH (*Smither et al., 2020*). Finally, studies using the Syrian hamster animal model have demonstrated the transmission of SARS-CoV-2 from infected animals to susceptible contacts through aerosolized SARS-CoV-2 (*Cimolai, 2020*; *Boon et al., 2022*). Results suggestive of aerosol transmission of SARS-CoV-2 also have been obtained using the ferret model (*Kim et al., 2020*; *Cimolai, 2020*).

Arguments against the importance of aerosol transmission of viruses have included mention of the paucity of clear data on detectable infectious viruses (as opposed to genomic material) in air samples. It is likely that this reflects the relatively high dilution of virus with time and distance from the contaminating source, the relative insensitivities of the viral infectivity assays, compared with the nucleic acid testing approaches more typically employed for detecting virus in air samples, and the actual air sampling technologies that have been used (*Lednicky et al., 2020*; *Lednicky et al., 2021*; *Borges et al., 2021*; *Dinoi et al., 2022*; *Morawska & Cao, 2020*; *Ram et al., 2021*; *Correia et al., 2022*; *Hadavi et al., 2022*; *Silva et al., 2022*). Unfortunately, the detection of viral RNA in air samples does not provide convincing evidence of the infectivity of such aerosols. Methods for detecting infectious viruses in samples collected from air need to be optimized and standardized (*Silva et al., 2022*). In addition, it must be realized that cell culture assays used to detect infectious virus are very inefficient, especially for naturally occurring viruses which have not been adapted to propagate in continuous cell lines. Ten percent or less of infectious viruses may be detected in cell culture (*Ward, Knowlton & Pierce, 1984*; *Mahalanabis et al., 2010*). A survey of available evidence for recovery of infectious respiratory virus from field air samples is shown in Table 1. The difficulties in detecting infectious viruses in air samples should not be used as evidence for the lack of presence of infectious virus in aerosols. As has been elegantly and succinctly stated: "The fact that there are no simple methods for

**Table 1  Studies reporting recovery of infectious respiratory virus from air samples in field studies.**

| Virus | Setting | Host Cell | Sampler used | RNA load[*] | Positive recoveries | Comment | Reference |
|---|---|---|---|---|---|---|---|
| Smallpox virus | Healthcare (India) | Chicken egg | Porton  Settling plates | NP | 5 of 47  12 of 30 | Air sampled from within 12″of patient's mouth | *Downie et al. (1965)* |
| SARS-CoV-2 | Healthcare (USA) | LLC-MK2, Vero E6 | BioSpot VIVAS BSS300PVIVAS prototype | Ct values ranged from 36.00 to 38.69 | 4 of 4 | Air samples from within 2 m of the patients, with 72 h and $\geq$96 h duration of symptoms. | *Lednicky et al. (2020)* |
| SARS-CoV-2 | Automobile (USA) | Vero E6 | Sioutas PCIS | Ct value of 33.50 | 1 of 5 | Driver of the car had 48 h symptomology when sampling occurred | *Lednicky et al. (2021)* |
| SARS-CoV-2 | Laboratory (USA) | Vero E6-TMPRSS2, A549-ACE2 | Gesundheit-II | >$10^4$ GC, <75 GC | 2 of 100 | Samples of exhaled air from COVID-19 cases with early mild infection | *Adenaiye et al. (2022)* |
| SARS-CoV-2 | Residential bedroom (USA) | LLC-MK2, Vero E6 | BioSpot VI-VAS 300, NIOSH Model BC-251 | $2.3 \times 10^6$ GE, $1.0 \times 10^6$ GE | 2 of 9 | Bedroom air samples for a COVID-19 patient | *Vass et al. (2022)* |
| SARS-CoV-2 | Laboratory USA) | Vero E6-TMPRSS2, A549-ACE2 | Gesundheit-II | $3.0 \times 10^4$ GC, $3.6 \times 10^2$ GC, $1.8 \times 10^7$ GC, $2.9 \times 10^3$ GC | 4 of 62 | Samples of exhaled air from COVID-19 cases infected with Delata and Omicron variants | *Lai et al. (2023)* |
| SARS-CoV-2 | Hospital Rooms (Canada) | Vero E6 | Liquid Spot Sampler Series 110A, SKC Eighty Four cassettes | Ct value of 32 | 1 of 4 | Air samples collected from hospital rooms of COVID-19 patients | *Fortin et al. (2023)* |

**Notes.**

[*]RNA load for culture-positive samples.

Ct, real-time PCR cycle threshold; GC, genomic copies; GE, genomic equivalents; NP, not performed.

Additional abbreviations used: h, hours; SARS-CoV-2, severe acute respiratory syndrome coronavirus 2.

detecting the virus [SARS-CoV-2] in air does not mean that the viruses do not travel in air." (*Morawska & Cao, 2020*).

Clinical evidence for the relevance of the aerosol route of transmission of SARS-CoV-2 has been reviewed extensively (*e.g., Tang et al., 2020*; *Zhang et al., 2020*; *Zhang & Duchaine, 2021*).

## Short- to long-range, long-time dissemination *via* contaminated environmental surfaces

The WHO assessment of the role of fomites in SARS-CoV-2 transmission is encapsulated in the following: "Despite consistent evidence as to SARS-CoV-2 contamination of surfaces and the survival of the virus on certain surfaces, there are no specific reports which have directly demonstrated fomite transmission. People who come into contact with potentially infectious surfaces often also have close contact with the infectious person, making the distinction between respiratory droplet and fomite transmission difficult to discern. However, fomite transmission is considered a likely mode of transmission for SARS-CoV-2, given consistent findings about environmental contamination in the vicinity of infected cases and the fact that other coronaviruses and respiratory viruses can transmit this way." (*World Health Organization, 2020*). This seems an odd way to downplay the risk associated with indirect transmission, since the opposite stance is equally applicable, *i.e.,* people who have close contact with the infectious person also may come into contact with potentially infectious surfaces, making the distinction between fomite and respiratory droplet transmission difficult to discern! This same type of argument is commonly used to dismiss the importance of indirect transmission, as "In the few cases where direct contact or fomite transmission is presumed, respiratory transmission has not been completely excluded." (*Meyerowitz et al., 2021*).

In the categorization of *Li (2020)* and *Marr & Tang (2021)*, this transmission mode corresponds to "surface" or "touch" transmission and can involve contaminated animate or inanimate surfaces (fomites). This pathway typically is considered short to long range and long time. Arguments for the importance of the indirect route of respiratory viral transmission include the known persistence of infectivity of these viruses once deposited on environmental surfaces (*Bean et al., 1982*; *Duan et al., 2003*; *Wolff et al., 2005*; *Otter et al., 2016*; *Bonny, Yezli & Lednicky, 2018*; *Kraay et al., 2018*; *Aboubakr, Sharafeldin & Goyal, 2020*; *Bueckert et al., 2020*; *Chin et al., 2020*; *Fong et al., 2020*; *Harbourt et al., 2020*; *Kampf et al., 2020b*; *Ren et al., 2020*; *van Doremalen et al., 2020*; *Castaño et al., 2021*; *Ijaz et al., 2021c*; *Johnson et al., 2021*; *Morris et al., 2021*; *Geng & Wang, 2022*; *Hadavi et al., 2022*). Numerous studies have reported the detection of viral RNA from environmental surfaces in the vicinity of patients (*e.g., Boone & Gerba, 2005*; *Chia et al., 2020*; *Dargahi et al., 2021*; *Dumont-Leblond et al., 2021*; *Kotwa et al., 2021*; *Maestre et al., 2021*; *Geng & Wang, 2022*; *Hadavi et al., 2022*; *Lin et al., 2022*; *Onakpoya et al., 2022*; *Shankar et al., 2022*) while relatively few studies have described the recovery of  infectious  respiratory virus on environmental surfaces (*Bonny, Yezli & Lednicky, 2018*; *Fong et al., 2020*; *Kotwa et al., 2021*; *Lin et al., 2022*). A survey of available evidence for recovery of infectious respiratory virus from naturally contaminated fomites in the field is shown in Table 2. Again, detection

of viral RNA on an environmental surface does not demonstrate the presence of infectious virus on that surface. In fact, this has been used, in part, as an argument against the importance of the indirect pathway (*Goldman, 2021*). In nucleic acid detection assays, there is an inverse relationship between observed Ct value and amount of nucleic acid detected, such that lower Ct values reflect greater amounts of nucleic acid in the sample. A systematic review of cell-based infectivity assay studies for assessing the presence of infectious SARS-CoV-2 virus on environmental surfaces has been published by *Onakpoya et al. (2022)*. These authors reported that isolation of infectious virus from fomites was significantly more likely in cases where genomic RNA recovery from those surfaces was associated with threshold cycle (Ct) values <30. This conclusion is reasonable, and has been reached by other researchers (*e.g.*, *Rando et al., 2021*; *Lin et al., 2022*), although the studies reviewed in Table 2 indicate that in some cases detection of viable virus has not been found to correlate with Ct number. As with air sampling technologies mentioned above, the relative inefficiencies in surface sampling technologies may play a part in the relatively low infectious virus recoveries observed in these studies. In addition, replication of wild type (naturally occurring) viruses is much less efficient than laboratory strains in cell culture, resulting in an underestimation of the quantity of infectious virus that might be present (*Ward, Knowlton & Pierce, 1984*). The risk significance of RNA isolation results for fomite contamination by SARS-CoV-2 and other respiratory viruses therefore remains to be determined. The inconsistencies of cell-based infectivity assays to confirm presence of infectious virus on surfaces displaying viral RNA may be a result of the insensitivities of the cell lines for infection by the target virus, or the relative inabilities of the cell lines to efficiently detect wild-type virus strains.

The same environmental factors that favor infectious viral survival in aerosols may also favor survival of infectious viruses on fomites. These include the specific matrices in which the viruses are deposited and the prevailing RH and temperature of the air indoor (*Otter et al., 2016*). Another consideration in terms of risk of indirect (fomite) transmission is the much longer persistence of infectivity of viruses on fomites than in the air (*Smither et al., 2020*; *Harbourt et al., 2020*). For example, while SARS-CoV-2 remains infectious only for minutes to hours in aerosols, it remains infectious on fomites for hours to days (*Harbourt et al., 2020*; *Smither et al., 2020*). Thus, contamination of fomites represents a longer-term risk of infection that should be taken into consideration when comparing relative risks of dissemination *via* aerosols *vs.* fomites.

Indoor environmental surfaces may become contaminated directly (*i.e.,* by expired respiratory droplets or other secretions/excretions of an infected person) or indirectly (*i.e.,* by transfer from an infected person's contaminated hands) (*Geng & Wang, 2022*). There is, therefore, a continual redistribution of pathogens occurring at the air-surface-hand nexus, each contributing to potential for viral dissemination (*Lei et al., 2017*; *Stephens et al., 2019*; *Scott et al., 2020*; *Scott, Bruning & Ijaz, 2021*).

Reports of the disruption of the chain of infection through targeted surface hygiene represent an indication of the possible relevance of the indirect transmission pathway. An example is a description of the reduction of norovirus contamination following use of targeted hygiene (*Barker, Vipond & Bloomfield, 2004*). Literature reviews of such

**Table 2  Studies reporting recovery of infectious respiratory virus from naturally contaminated fomite samples.**

| Virus | Sampling setting | Host Cell | Fomite sampled | Ct[*] | Positive Recoveries | Comments | Reference |
|---|---|---|---|---|---|---|---|
| HCoV-229E | Classroom (USA) | A549, MRC-5, Vero E6 | Desk top | NP | 4 of 8 | No patient data. Surfaces disinfected daily | *Bonny, Yezli & Lednicky (2018)* |
| | | | Door knob | NP | 2 of 4 | | |
| MERS-CoV | Healthcare (South Korea) | Vero E6 | Bed sheet | No correspondence between Ct and infectious virus isolation was reported | 1 of 15 | | *Bin et al. (2016)* |
| | | | Bed rail | | 1 of 15 | | |
| | | | IV fluid hanger | | 2 of 14 | | |
| | | | Computed radiography cassette | | 1 of 1 | | |
| | | | Anteroom table | | 1 of 7 | | |
| SARS-CoV-2 | Healthcare (Canada) | Vero | Used facial tissues | 5 to ~32 (Ct ≤ 25 is 84% predictive of infectious virus isolation) | 4 of 5 | Positive by RNA and by culture optimal within 1 week of onset of symptoms | *Lin et al. (2022)* |
| | | | Phone/call bell | | 2 of 10 | | |
| | | | Nasal prongs | | 3 of 4 | | |
| SARS-CoV-2 | Residential (USA) | LLC-MK2, Vero E6 | Mobile phone | $2.29 \times 10^4$ GE | 1 of 3 | Mobile phone of COVID-19 patient | *Vass et al. (2023)* |
| Influenza A and B virus | Healthcare (Hong Kong) | MDCK | Cell phone | No correspondence between Ct and infectious virus isolation was reported | 1 of 15 (Influenza A) 2 of 8 (Influenza B) | Time from symptom onset to sampling <72 h | *Xiao et al. (2021)* |
| Monkeypox virus | Residence (USA) | BSC-40 | Paper towels on bed | Infectious virus isolates corresponded to Ct values <24. | 1 of 1 | Sampling occurred 15 days after the infected person left the residence. Infectious virus was isolated more often from porous than non-porous surfaces. | *Morgan et al. (2021)* |
| | | | Underwear on bed | | 2 of 2 | | |
| | | | Blanket on couch | | 1 of 1 | | |
| | | | Towel on couch | | 1 of 1 | | |
| | | | Mattress cover in hamper | | 1 of 1 | | |
| | | | Coffee tabletop | | 1 of 1 | | |
| Smallpox virus | Healthcare (India) | Chicken eggs | Bed pillow | NP | 41 of 67 | Patients in acute illness | *Downie et al. (1965)* |
| | | | Bed sheet | | 15 of 16 | | |

**Notes.**

[*]Abbreviations used Ct, real-time PCR cycle threshold; GE, genome equivalents; HCoV-229E, human coronavirus 229E; MERS-CoV, Middle East respiratory syndrome coronavirus; NP, not performed; SARS-CoV-2, severe acute respiratory syndrome coronavirus 2.

studies performed to address the possible relevance of the indirect transmission route have been published (*Boone & Gerba, 2007*; *Otter et al., 2016*; *Stephens et al., 2019*; *Bueckert et al., 2020*; *Kampf et al., 2020a*; *Castaño et al., 2021*; *Chen, 2021*; *Chen, Jia & Han, 2021*; *Mohamadi et al., 2021*; *Dargahi et al., 2021*).

The ability of bacteriophage deposited on hands to be transferred to environmental surfaces has been demonstrated previously (*e.g.*, *Castaño et al., 2021*). Transferability of SARS-CoV-2 from environmental surfaces to skin (albeit pig skin), previously noted as a significant knowledge gap for the indirect pathway (*Ijaz, Nims & McKinney, 2021a*), has now been empirically demonstrated (*Johnson et al., 2021*). Transfer of virus from hands to environmental surfaces, and transfer of this respiratory virus from contaminated surfaces back to hands, has been demonstrated for rhinovirus (*Ansari et al., 1991*; *Winther et al., 2007*; *Kraay et al., 2018*), SARS-CoV-2 (*Gerba et al., 2023*), and for influenza virus and norovirus (*Kraay et al., 2018*). Similar results have been described for influenza A and B viruses (*Bean et al., 1982*). This topic has also been approached from the standpoint of interruption of transfer of infection through use of hand antisepsis (*Bidawid et al., 2004*) or reduction of hand and environmental surface burdens of bacteriophage following use of sanitizers (*Kurgat et al., 2019*).

## Re-aerosolization of virus from environmental surfaces

Re-aerosolization (also referred to as resuspension) of viruses from environmental surfaces is not viewed as an additional transmission mode, but rather is a contributing factor for aerosol virus burden. Re-aerosolization, therefore, represents an interdependency between droplet transmission, environmental surface contamination, and aerosol transmission of infectious respiratory viruses. The topic of virus re-aerosolization from environmental surfaces is relatively novel, and not surprisingly, the WHO has not—to our knowledge—acknowledged the possibility or possible relevance of re-aerosolization of virus from environmental surfaces in its website addressing transmission pathways (*World Health Organization, 2020*). What do we know about re-aerosolization, and why should we care? As alluded to previously, we know that viruses can remain infectious to varying degrees once deposited on environmental surfaces. Depending on the specific type of surface contaminated (inanimate (hard, soft) or animate), and upon the types of human activities (human walking, vacuuming, *etc.*) taking place on or around such surfaces once contaminated, there is a possibility of re-aerosolizing the virus. We have depicted some of the possibilities for pathogen re-aerosolization in Fig. 3.

Ironically, as depicted in Fig. 3, cleaning activities in general and, in particular, cleaning of flooring (carpeted or hard surface) and bathroom surfaces, are among the activities which can lead to re-aerosolization of viruses from contaminated environmental surfaces (*Sattar & Ijaz, 1987*; *Ijaz et al., 2016*; *Ciofi-Silva et al., 2019*; *Tang et al., 2020*; *Samet et al., 2021*; *Goforth et al., 2023*; C. Gerba, 2023, unpublished). Other types of human activities that may promote re-aerosolization of virus include walking, door opening, vacuuming, toilet flushing, moving curtains, and pretty much any activity that results in a vertical or horizontal air current (*Tang et al., 2020*; *Goforth et al., 2023*). Once re-aerosolized, infectious viruses and other pathogens form a vertical gradient, with higher concentrations

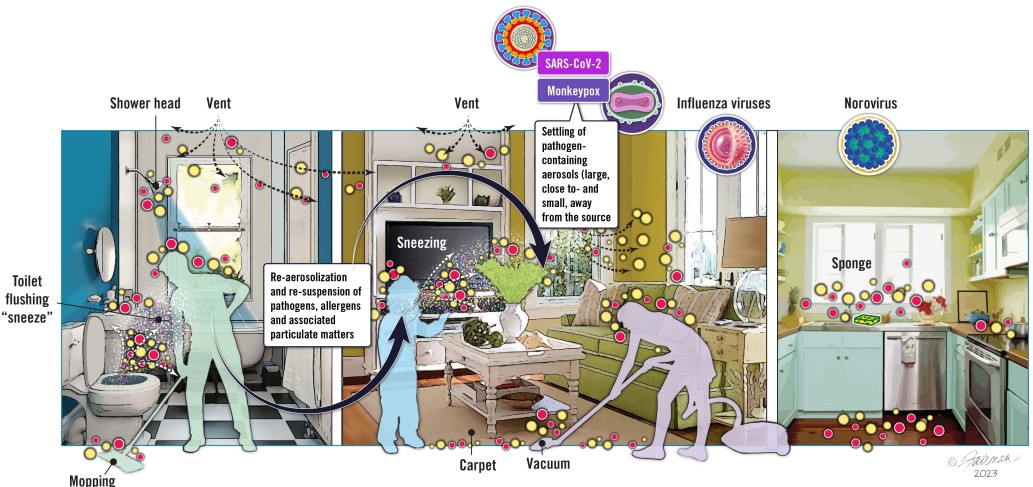

**Figure 3** **Schematic depiction of the contamination of fomites by infectious virus, showing possible opportunities for re-aerosolization of viruses.** Image source: ©Fairman Studios, LLC, CC-BY-ND-NC 4.0, https://creativecommons.org/licenses/by-nc-nd/4.0/deed.en.

being produced closer to the contaminated surface and concentrations being lower as the distance from the infectious aerosol source increases (*Khare & Marr, 2015*).

There is also the possibility of re-aerosolization of virus from contaminated personal protective garments when removing these, and from face mask materials that have been contaminated with virus (*Liu et al., 2020*). In addition, the potential for resuspension (re-aerosolization) of virus from contaminated fabric has been investigated (*Licina & Nazaroff, 2018*; *Kvasnicka et al., 2022*). The latter raises the very concerning possibility that replacing bedsheets *etc.* from patient beds (under both home, community and healthcare settings) and removal of protective equipment itself may represent a source of viral aerosols and, therefore, a route for dissemination of such viruses from infected individuals to healthcare workers in patient care settings.

Once re-aerosolization of a virus occurs, the aerosol likely migrates within air currents, but may also "hitchhike" (associate with, and be carried by) the various types of aerial particulate matter (*e.g.*, dust, allergens, PM) to be discussed in the next section of this review. Therefore, it needs to be emphasized that decontamination of environmental surfaces also reduces infectious droplets/aerosols by reducing fomites available for re-aerosolization through various activities as noted above (*Leung, 2021*).

### Long-range and time dissemination via aerial particulates

One of the more novel interdependencies associated with transmission of respiratory viruses includes the potential for facilitation of long-range transmission of respiratory aerosols by aerial particulates (buoyant aerial particulate matter) (*Chen, Jia & Han, 2021*; *Gu et al., 2023*). That is, there is at least a theoretical possibility that such viruses might be conveyed by aerial particulates including, but likely not limited to, air pollutants ($PM_1$, $PM_{2.5}$, $PM_{10}$), dust, and allergens (*Asadi et al., 2020*; *Barakat, Muylens & Su, 2020*;

*Bashir et al., 2020*; *Comunian et al., 2020*; *Farhangrazi et al., 2020*; *Setti et al., 2020*; *Tung et al., 2021*; *Betsch & Sprengholz, 2021*; *Brandt & Mersha, 2021*; *Cao et al., 2021a*; *Chen, Jia & Han, 2021*; *Santurtún et al., 2022*; *Solimini et al., 2021*; *Tchicaya et al., 2021*; *Woodby, Arnold & Valacchi, 2021*; *Yilbas et al., 2021*; *Baron, 2022*; *Tao et al., 2022*). As of this time, the possibility of airborne particulates as possible carriers of SARS-CoV-2 or other respiratory viruses, has not been discussed in the WHO guidance (*World Health Organization, 2020*).

It is very unlikely that free infectious virus/monodispersed particles are expelled from infected persons in quantities relevant for transmission. It seems more likely that virus is associated with body fluids/respiratory secretions or other physiological substrates (*Zhang & Duchaine, 2021*) (Figs. 1 and 3). Solvated viral particles may subsequently associate with airborne dust, allergens, and air pollutants such as fine particulate matter with an aerodynamic diameter < 2.5 $\mu$m (referred to as $PM_{2.5}$). In fact, infectious respiratory droplets, droplet nuclei, and aerosols might be thought of as relatively transitory states, with conveyance on aerial particulates representing a condition leading to greater airborne distribution, and possibly stability (*Zuo et al., 2013*). Each of the categories of aerial particulate matter mentioned above may contribute to long-range transmission of SARS-CoV-2 and other respiratory viruses. Furthermore, the physical and chemical properties of the airborne particulates with which the viruses associate (size, chemical constituents, electrostatic charge) may favor virus adsorption and survival (*Gu et al., 2023*) as well as act synergistically with the virus through change in tissue expression of viral receptors and entry factors in humans exposed to the particulates (see 'Discussion' below), thereby enhancing susceptibility to infection in these infectious virus-exposed individuals.

The earliest clues of the possibility of SARS-CoV-2 conveyance on aerial particulate matter have been associations between COVID-19 case incidence and levels of airborne $PM_{10}$ and $PM_{2.5}$ in China, France, Italy, the UK, and The Netherlands (*Barakat, Muylens & Su, 2020*; *Farhangrazi et al., 2020*; *Chen, Jia & Han, 2021*; *Cao et al., 2021a*; *Comunian et al., 2020*; *Santurtún et al., 2022*; *Tchicaya et al., 2021*; *Tung et al., 2021*; *Zhang & Duchaine, 2021*). An association between COVID-19 cases and case mortality and $PM_{2.5}$ and $PM_{10}$ levels was reported also in a study conducted in California (*Bashir et al., 2020*). Detection of SARS-CoV-2 associated with PM has, unfortunately, relied on RNA measurement, not isolation of infectious virus (*Setti et al., 2020*; *Cao et al., 2021a*; *Santurtún et al., 2022*; *Tung et al., 2021*). Confusion exists in the literature on this point, as the statement "viable avian influenza viral RNA was found in PM…" (*Tung et al., 2021*) indicates. Certain assumptions have been made in promoting the hypothesis of viral transmission on particulates: (1) that virus survival measured in aerosol studies informs the survival of virus attached to particulates; and (2) that virus adhered to particulates can initiate infections in humans, and animal models (such as the hamster). There do not appear to be any empirical data to support these two assumptions, which have been defended, therefore, on purely theoretical grounds (*Farhangrazi et al., 2020*).

There are some interesting implications if transmission *via* aerial particulate matter is found to be relevant to infection transmission. For instance, $PM_1$, $PM_{2.5}$, and $PM_{10}$, and cigarette smoke has been shown in animal studies to upregulate expression of the ACE2

receptor involved in entry of SARS-CoV-2 into host cells and this has been confirmed in *in vitro* studies involving human cell lines (*Lin et al., 2022*; *Cao et al., 2021a*; *Tung et al., 2021*). In addition, $PM_{2.5}$ has been shown to upregulate the expression of transmembrane serine protease 2 (Tmprss2) in animal studies (*Cao et al., 2021a*; *Kaur et al., 2021*). This protease is thought to cleave spike protein as a necessary step in the process of SARS-CoV-2 entry into the cell (*Hoffmann et al., 2020*; *Rando et al., 2021*). There is, therefore, a theoretical possibility that $PM_{2.5}$ might serve as a vehicle for long-range transmission of SARS-CoV-2, and convincing evidence that this carrier upregulates proteins in the host that might enhance susceptibility to infection.

## Is there a definitive conclusion to the transmission debate?

The debate over the relative importance of aerosol *vs.* droplet transmission (*i.e.,* short-range *vs.* long-range transmission) has been difficult to resolve in the face of knowledge gaps that have yet to be closed. What data do we need in order to resolve the debate over the relative significance of transmission *via* infectious respiratory droplets *vs.* aerosols? Is this more a debate over short- *vs.* long-range transmission? What roles do indirect transmission and re-aerosolization of viruses play, and does viral conveyance on aerial particulate matter really play a significant role in aerosol transmission and acquisition or severity of infection?

It has been suggested that droplets and aerosols should just be viewed as extremes in a size spectrum, and what really counts is the ability to deliver the infectious viral payload to the most susceptible site (*i.e.,* the upper respiratory mucosa) (*Zhang & Duchaine, 2021*; *Ram et al., 2021*). In any case, we have several knowledge gaps to resolve (*Leung, 2021*; *Wang et al., 2021*). For instance, we do not have good empirical data on the quantity of *infectious* virus that typically is released from infected individuals in the form of respiratory droplets or aerosols, and this will depend on the status of the host and the stage of the infection (asymptomatic, pre-symptomatic, symptomatic, post-symptomatic). Specifically, how many infectious virus particles (not RNA!) are released from the infected person as a result of breathing, talking, coughing, sneezing, or mechanical activities, *e.g.*, vacuuming floor/carpet, toilet flushing, *etc.*? The extent to which the released infectious viruses end up in respiratory droplets or aerosols could be inferred from the data on size distribution of droplets expelled during different expiratory activities (*Morawska et al., 2009*), but this needs to be verified experimentally. The numbers of infectious SARS-CoV-2 virions required to initiate an infection in individuals may depend on a number of factors, including susceptibilities of the individuals exposed. Recent papers have suggested that relatively low numbers of infectious viruses (*e.g.*, as low as 10 $TCID_{50}$ of a wild-type virus (SARS-CoV-2/human/GBR/484861/2020)) are required, as reported for volunteers without previous SARS-CoV-2 infection or vaccination (*Killingley et al., 2022*). The relatively low infectious dose for SARS-CoV-2 has been observed also in clinical and modeling studies (*Prentiss, Chu & Berggren, 2022*; *Stettler et al., 2022*). More clarity as to the quantities of infectious virus (not viral RNA) carried in aerosols (including aerial particulate matter) and droplets, or capable of being conveyed to viral-permissive mucous membranes *via* hands after touching contaminated environmental surfaces, may help inform the relevance of the various direct and indirect pathways for infection transmission (Fig. 1). However, in

view of the relatively low $TCID_{50}$ for SARS-CoV-2, it appears that the aerosol- and fomite transmission pathways could indeed be relevant.

The WHO has made the point (*World Health Organization, 2020*) that it is difficult to separate potential direct and indirect exposure modes in establishing transmission relevancy, and this difficulty contributes greatly to the debate mentioned above. Bringing infected and non-infected individuals into close proximity increases the opportunities for contamination of air with infectious droplets and with infectious aerosols, and for the contamination of environmental surfaces within the occupied space indoors. It, therefore, is difficult to dissect out the most important factors for infectious virus transmission. The relatively great difficulties in establishing the levels of infectious virus in air samples taken in the vicinity of COVID-19 patients further confounds the debate (*Zhang & Duchaine, 2021*; *Meyerowitz et al., 2021*). We have previously identified current knowledge gaps regarding the relevance of the indirect transmission route (*Ijaz, Nims & McKinney, 2021a*), in order to encourage further research toward resolving these gaps. Common to both the direct and indirect routes is the question of exactly how much infectious virus must reach susceptible oral, nasal, or ocular mucous membranes or the lung to lead to initiation of infection. The human infectious dose leading to infection of 50% of those exposed ($ID_{50}$) may depend upon the infection initiation site (most likely the upper respiratory tract per *Lamers & Haagmans, 2022*), although infection initiation in the ocular mucous membranes and the lower respiratory tract is also possible. Particular viral infection initiation sites may render the infected person more vulnerable to acquiring COVID-19 or may affect the severity of the acquired disease. The age and immune status of the host, and other pre-existing conditions (*e.g.*, diabetes, asthma, cardiac issues, *etc.*) may determine the clinical outcome post-infection. Certain aerial particulates can act synergistically to increase likelihood of acquiring an infection (*e.g.*, by promoting ACE-2 receptor or protease expression in the case of SARS-CoV-2), or influencing the clinical severity of the acquired infection. These represent topical areas deserving of further study. Resolving these knowledge gaps may help investigators to determine the relative relevance of the various transmission pathways discussed in this article.

It is believed by many that long-range transmission by aerosols will be proved only by first excluding the possibility of droplet transmission. Flipping this argument, can we make conclusions about the importance of the close-range droplets transmission mode without first ruling out the possibility of long-range aerosol transmission (*Leung, 2021*) or of indirect transmission (*Ijaz, Nims & McKinney, 2021a*)? The relatively low reproduction number for SARS-CoV-2 (~2.5, *vs.* ~18 for measles virus) has been used as an argument against an important contribution of long-range aerosol transmission (*Klompas, Baker & Rhee, 2020*) for SARS-CoV-2. The various variants of concern for SARS-CoV-2 have been reported to exhibit similarly low reproduction numbers (the highest reproduction values being 1.22 for Alpha, 1.19 for Beta; 1.21 for Gamma, 1.38 for Delta, and 1.90 for Omicron) (*Manathunga, Abeyagunawardena & Dharmaratne, 2023*). The reproduction number for SARS-CoV-2 aligns with those for other respiratory viruses, including SARS-CoV and the influenza virus causing the 1918 pandemic (2.0–3.0), MERS-CoV (0.9), and the H1N1 influenza A virus causing the 2009 pandemic (1.5) (*Peterson et al., 2020*). There may be

**Table 3  Key points for various transmission modes and potential interdependencies.**

| Transmission mode | Sources | Range (time)[b] | Consequences and interdependencies |
|---|---|---|---|
| Infectious respiratory droplets (drop spray transmission)[*] | Talking, coughing, sneezing, etc. | Short (Short) | • Direct transmission to susceptible individuals' upper respiratory tract/oral and ocular mucosa<br>• Infectious droplet dehydration to contribute to aerosol transmission<br>• Infectious droplet deposition onto environmental surfaces[a], leading to indirect transmission |
| Infectious viral aerosols (aerosol inhalation transmission)[*] | Talking, coughing, sneezing, droplet nuclei, toilet flushing, other aerosol-generating procedures | Short to Long (Short to Long) | • Direct transmission to susceptible individuals' upper and lower respiratory tract and oral and ocular mucosa<br>• Conveyance via aerial particulates, leading to direct transmission to susceptible individuals' upper respiratory tract and oral and ocular mucosa and possible synergistic effects |
| Contaminated fomites (surface or touch transmission)[*] | Droplet deposition, other excreta from infected individuals | Short to long (Long) | • Indirect transmission to susceptible individuals' upper respiratory tract/ocular/oral mucosa via hands<br>• Re-aerosolization contributing to aerosol transmission |

**Notes.**

[*]A proposed new terminology (*Li, 2020*; *Marr & Tang, 2021*).

[a]Infectious virus-environmental surfaces.

[b]See Fig. 2 for more detail on the distance ranges and times for the various transmission modes. Exceptions to these categorizations may exist, with small to intermediate droplets acting more like aerosols with regard to distances covered and areas of the respiratory tract impacted.

other reasons for the low reproduction number for respiratory viruses including SARS-CoV, influenza virus, and SARS-CoV-2. For instance, it has been proposed (*Tong, 2007*) that the low reproduction number for SARS-CoV is due in part to individual-to-individual differences in susceptibility to acquisition of the virus (*i.e.,* heterogeneity of individual infectiousness). This heterogeneity in the case of SARS-CoV and SARS-CoV-2 might relate to the possible synergistic effects of modifiers of host cell receptor or protease expression, rather than to mode of transmission. Until this is worked out, a reasonable position to take is that the transmission of SARS-CoV-2 is multimodal (*Brosseau, 2020*; *Klompas, Baker & Rhee, 2020*; *Ijaz, Nims & McKinney, 2021a*; *Leung, 2021*), including direct physical contact, infectious droplet exposure, infectious aerosol exposure, and indirect transmission through contaminated environmental surfaces (Table 3). It seems to us that to focus on only one of the possible modes of viral transmission is risky, from a public health point of view, particularly if mitigation approaches intended for IPAC are informed by this simplistic approach. Other investigators appear to share this view, as well (*e.g., Killingley & Nguyen-Van-Tran, 2013*; *Tang et al., 2020*; *Chen, 2021*; *Leung, 2021*; *Lin et al., 2022*).

In two of the few studies evaluating the possible presence of *infectious* virus in air samples, it has been reported that infectious SARS-CoV-2 was isolated from air samples collected 2 to 4.8 m away from COVID-19 patients (*Lednicky et al., 2020*), or in air samples taken from the interior of a car driven by a COVID-19 patient (*Lednicky et al., 2021*). While infectious virus was detected in the car air sampling study, quantitation of the virus concentration was not possible. The levels detected in the hospital room study were 6 to 74 tissue culture infectious dose$_{50}$ (TCID$_{50}$) per liter of air sampled. This level of virus was

sufficient to cause infection in the Syrian hamster animal model (*Boon et al., 2022*) and may also be relevant for human infection (*Killingley et al., 2022*; *Prentiss, Chu & Berggren, 2022*; *Stettler et al., 2022*).

## Approaches for preventing dissemination of respiratory viruses

During the SARS-CoV-2 (COVID-19) pandemic, health authorities worldwide responded by implementing a variety of pharmaceutical (*i.e.,* vaccines, therapeutics) and non-pharmaceutical interventions (NPI) to mitigate the risks associated with the illness and risk of transmitting the virus to others (*Ijaz et al., 2022*). The NPI were most typically implemented as groups of interventions based on consideration of most of the transmission modalities discussed in this paper. The types of NPI have included face mask wearing mandates, increased hand hygiene, air purification/sanitization, and social distancing measures, testing and isolation of virus-positive individuals, school and business/entertainment closures, intra- and inter-country travel restrictions, bans on indoor or outdoor social gatherings, and stay-at-home (shelter-in-place) mandates (*Ijaz et al., 2022*). That these interventions have been successful has been indicated not only by decreases in new caseloads for SARS-CoV-2, but also by the profound decreases in caseloads observed for seasonal respiratory infections such as respiratory syncytial virus, influenza virus, human metapneumovirus, *etc.* (*Ijaz et al., 2022*).

Going forward, we advocate taking the most conservative approach informed by the results of the present review, that is, in a confined indoor space occupied both by infected individual(s) and susceptible individual(s) including those immunocompromised, the risk of exposure to infectious respiratory viruses may include infectious respiratory droplets and infectious aerosols of varying size, indirect transmission from environmental surfaces, and virus re-aerosolized from previously contaminated environmental surfaces (Table 3). Under certain circumstances, risk of respiratory virus conveyed from other indoor or outdoor locations by aerial particulates (environmental pollutants, allergens, dust, *etc.*) should also be acknowledged. In this section, we discuss available means of mitigating these transmission risks.

### Personal protective equipment

The most commonly employed NPI for mitigating risk of acquiring a respiratory viral infection COVID-19 pandemic worldwide, has been universal use of face masks. Countries which responded to the pandemic with universal mask wearing mandates (a good example being Taiwan) have been most effective in reducing virus spread (*Prather, Wang & Schooley, 2020*). A properly fitted face mask is especially important for limiting the spread of virus from asymptomatic or pre-symptomatic infected individuals, while being essential in the case of symptomatic individuals (*Bard et al., 2019*; *Prather, Wang & Schooley, 2020*).

### Air purification/sanitization technologies

Air decontamination within indoor spaces is achievable using a variety of off-the-shelf engineering applications (*Ijaz et al., 2016*; *Morawska et al., 2020*; *Sattar et al., 2016*; *Berry et al., 2022*; *Tang et al., 2020*; *Burdack-Freitag et al., 2022*). Recommendations for air purification/sanitization in indoor spaces should be considered, especially in higher risk

situations, such as crowded indoor spaces in public (in-home social event, community, transportation *etc.*) settings, healthcare settings, and in the presence of an individual known to be infected with a respiratory virus, *etc.* (*Samet et al., 2021*; *Ijaz et al., 2022*). Air purification/sanitization approaches range from simply increasing the number of air changes per hour (*Kormuth et al., 2018*; *Tang et al., 2020*; *Li, 2021*; *Rencken et al., 2021*; *Cheng et al., 2022*), use of hydrogen peroxide fogging (*Hislop, Grinstead & Henneman, 2022*; *Lee & Henneman, 2022*), use of ozone plus high humidity sparging devices (*Dubuis et al., 2020*), use of high efficiency particulate air (HEPA) filtration, use of ultraviolet light devices or atmospheric plasma reactors installed within heating, ventilation and air conditioning (HVAC) systems (*Garcíade Abajo et al., 2020*; *Tang et al., 2020*; *Kormuth et al., 2018*; *Burdack-Freitag et al., 2022*; *Pourchez et al., 2022*), use of portable air purifying devices (*Bard et al., 2020*; *Rodríguez et al., 2021*; *Sattar et al., 2016*), and use of essential oil vaporizing devices (*Mirskaya & Agranovski, 2021*). In low- to medium-income regions of the world, widespread utilization of certain of these technologies may not be economically practical.

### Decontamination of environmental surfaces

For transmission occurring *via* the contaminated environmental surfaces/hands/susceptible mucous membrane nexus, it is clear that surface and hand hygiene are the primary risk mitigating approaches. For enveloped respiratory viruses such as coronaviruses, a variety of microbicides should have adequate efficacy to decrease environmental surface burdens of these viruses (*United States Environmental Protection Agency, 2020*). As the SARS-CoV-2 pandemic has lessened in severity, of course, there has been relaxation of NPI, such as social distancing and mask wearing, and a global resurgence of cases attributed to other respiratory viruses (*Ijaz et al., 2022*, and references therein). These have included illnesses caused by other enveloped viruses, such as influenza viruses, respiratory syncytial virus, parainfluenza virus, and human metapneumovirus. A variety of commonly employed microbicides should be useful for surface and hand hygiene in the such cases (*Larson et al., 2004*; *Wolff et al., 2005*; *United States Centers for Disease Control and Prevention, 2011*; *United States Centers for Disease Control and Prevention, 2020a*; *United States Centers for Disease Control and Prevention, 2020b*; *United States Centers for Disease Control and Prevention, 2020c*; *Golin, Choi & Ghahary, 2020*; *Kampf et al., 2020b*; *Lei et al., 2020*; *Castaño et al., 2021*; *Ijaz et al., 2021b*; *Ijaz et al., 2021c*; *Scott, Bruning & Ijaz, 2021*; *Cimolai, 2022*). However, not all respiratory viruses are enveloped, as the non-enveloped rhinoviruses and adenoviruses may also contribute to caseloads. For surface and hand hygiene of the latter viruses, more broad-spectrum active ingredients may be required (*Barker, Vipond & Bloomfield, 2004*; *Bidawid et al., 2004*; *Whitehead & McCue, 2010*).

Awareness of the possibility for re-aerosolization of virus from contaminated environmental surfaces should be considered when discussing mitigation approaches. Re-aerosolization effectively results in a switching of transmission routes from the indirect route (environmental surfaces -hand-mucous membrane nexus) to the airborne route (infectious aerosol transmission, transmission by aerial particulate matter). One might argue that prevention of re-aerosolization from environmental surfaces might be

adequately prevented through stringent surface hygiene practices, although the possibility exists that at least some re-aerosolization may occur because of cleaning practices (*e.g.*, vacuuming carpeted or hard surface flooring, moving contaminated curtains, *etc.* (*Ciofi-Silva et al., 2019*; *Tang et al., 2020*; *Samet et al., 2021*; *Goforth et al., 2023*; C. Gerba, 2023, unpublished).

## CONCLUSIONS

Although we now apparently are in the final stages of the ongoing SARS-CoV-2 pandemic, and the past three years have seen an explosion in research on all aspects of the pandemic, it is clear that we still have much to learn about transmission of SARS-CoV-2 and other respiratory viruses. New variants of SARS-CoV-2 will likely continue to emerge (*e.g.*, *World Health Organization, 2023*). It would appear to be beneficial to public health if individual and group biases regarding the perceived importance of the various transmission modalities could be set aside and a more holistic view encompassing the truly multimodal nature of respiratory virus dissemination could be acknowledged. This view is informed by the concept of a continuum of sizes for infectious droplets and infectious aerosols, the importance of residence time over which viruses remain airborne and infectious in droplets, aerosols, and particulate matter, and the reality of the many interdependencies observed between the varying transmission pathways. In a practical sense, the global responses to the SARS-CoV-2 pandemic correctly considered most of the modes of transmission discussed herein during implementation of regionally-mandated non-pharmaceutical interventions intended to limit viral dissemination. Messaging from health authorities could be improved going forward, to limit confusion about sources of risk. Until we have definitive data to rule out one or more of the transmission modalities, and considering the numerous interdependencies among the various modalities, we should not discount the importance of any specific modalities in our official messaging.

## ACKNOWLEDGEMENTS

We thank Dr. Chris Jones and Dr. Mark Ripley, both from Reckitt Benckiser R&D, for their critical review of the manuscript and feedback. The authors gratefully acknowledge Jennifer Fairman for creating the illustrations in Figs. 1 and 3.

### Funding

The authors received no funding for this work.

### Competing Interests

Julie McKinney and M. Khalid Ijaz are employed by Reckitt Benckiser LLC. Raymond W. Nims is employed by Syner-G BioPharma, and received a fee from Reckitt Benckiser LLC for his role in authoring and editing the manuscript. Reckitt Benckiser LLC participated in the decision to publish. No other competing interests are declared by the authors.

## Author Contributions

- M. Khalid Ijaz conceived and designed the review, performed the review, analyzed the data, prepared figures and/or tables, authored or reviewed drafts of the article, and approved the final draft.
- Syed A. Sattar analyzed the data, authored or reviewed drafts of the article, and approved the final draft.
- Raymond W. Nims conceived and designed the review, performed the review, analyzed the data, prepared figures and/or tables, authored or reviewed drafts of the article, and approved the final draft.
- Stephanie A. Boone analyzed the data, authored or reviewed drafts of the article, and approved the final draft.
- Julie McKinney analyzed the data, authored or reviewed drafts of the article, and approved the final draft.
- Charles P. Gerba analyzed the data, prepared figures and/or tables, authored or reviewed drafts of the article, and approved the final draft.

## Data Availability

This is a literature review.

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
