# Peer review of "Environmental dissemination of respiratory viruses: dynamic interdependencies of respiratory droplets, aerosols, aerial particulates, environmental surfaces, and contribution of viral re-aerosolization"

_PeerJ, doi:10.7717/peerj.16420_

## Round 0.1 · original submission · Minor Revisions

The review process is now complete, and two reviews are included at the bottom of this letter. All reviewers, with my own assessment, agree that your Literature Review deserves to be published. While the manuscript is well written and referenced, we identified some concerns that must be rigorously considered in your resubmission.

·

Basic reporting

Most of the prior literature was appropriately referenced, but some references were seen to lack DOIs, for example, the reviewer would like the authors to ensure all references are to the same standard throughout.

Experimental design

The survey methodology is within the scope of both the reviewer and what the authors describe to be a good standard, technically correct and has good coverage of the subject with respect to viral transmission (especially SARS-CoV-2). However, the review of mitigation at the end of the article (line 544 onwards) is more limited and doesn’t seem to add a great deal to an otherwise well-developed paper. Perhaps this could be either developed more or reduced in substance. For example, little space is given to face masks, probably one of the most important tools in preventing bacterial/viral spread, as compared to other devices or methods.

Validity of the findings

The multi-modal discussion in the conclusions has been reviewed well in the conclusion’s text. However, has the research questions posed in the introduction been fully reviewed in the conclusions? Do you need to redefine the transmission pathways and major arguments, this might be NO? Are there any other questions within the Intro that needing reviewing?

Perhaps the conclusion(s) could be more forthright with this messaging to those it is communicating to.

Do the authors think there are any gaps in the research on viral transmission, should this be stated in the conclusions?

Additional comments

Overall, I found this review article to be of great interest and engaging. These are suggestions, general comments or corrections for this draft.

27) “viral dissemination” – viral spread or transmission might be more concise or clearer at this point
28) “…and fomite spread, Depending on humidity…” - should this comma be a full stop?
44) Keywords “Aerial fomites” – is this a new phrase or wording and if so, it probably does require definition – upon searching in Google it wasn’t defined previously
332) (Otter et al.). - missing year
402) “Once re-aerosolization of a virus occurs, the aerosol may migrate with air currents, but more
likely hitchhikes…” – this is surmised and without foundation (or basis, unless backed by research or articles), is it more likely the after re-aerosolization and stay with the air currents than to hitchhike? Thus, I think this could be rewritten to reconsider the lack of evidence and could be written in a more neutral phrase. Also, does this communicate well with the authors’ stance “to take a neutral position”?
418) “It is inconceivable…” – Not considered particularly scientific language suggest use of perhaps ‘Very unlikely’
432) “Is this possibility a reality? What evidence do we have of SARS-CoV-2 transmission by deposited droplets-reduced-to-fomites, aerial fomites?” – Could be more formally written, this is a little casual in its manner, the language would suit being more scientific and it is suggested that this sentence be rewritten.
448) “so-called” – not sure I (readers will) understand what the authors are getting at here, though it is difficult to find an alternative phrase. Could this be reconsidered?
471) there are plenty of sources for infection data – try Stettler et.al. (2022) https://doi.org/10.1098/rsos.212022
480-482) Does the paragraph(s) from here answer the question “droplet and aerosol transmission is likely to produce a higher dose of virion” compared to “indirect transmission or aerial fomites” and is more likely to produce the ID50? I think it might, but is this clear to the reader? Ensure transmission relevancy is clear.
509) Do newer variants have higher transmission rates – could this be checked prior to publication?

Reviewer 2 ·

Basic reporting

The paper is well written, referenced and organised appropriately for the most part. Unfortunately one table is included but not referenced in the text.

The work is of broad interest and relevant for public health advice.

I do not believe that this element has been reviewed recently.

Experimental design

The methodology for the literature is informal (and so not entirely transparent). However, the review appears to be comprehensive in its coverage.

The review is organised into sections appropriately.

The review, although covering all mechanisms, is clearly aimed at exploring the role of resuspended fomite-bound virus. Having said that it doesn't appear to be subject to bias.

Validity of the findings

The review does highlight a potentially neglected mechanism. There are a number of technical comments highlighted below in Section 4. Additional comments that have some bearing on the validity. The work does set out unresolved questions that should be addressed.

Additional comments

Major comments
p5 - Given the commercial affiliations of some of the authors the reader may assume some vested interest in emphasising certain transmission mechanisms over others. It would be helpful to have a statement of interests to avoid the work being discounted on that basis.

p7, l28-31 - I think the reality isn't so much a case of do they evaporate or grow, more how quickly do they evaporate, for most environments. The range of sizes produced means that many droplets will deposit even if they dry. The initial droplet size range that remains airborne will vary depending on the conditions. This sentence oversimplifies things and makes a complex process a binary one unnecessarily.

p8, l67 - This sentence implies that the only transport process for droplets is via resuspension from fomites. I don't think this is the authors' intention and the final clause about dispersion in turbulent air jets highlights another transport mechanism. I think the precision of the words is important here though. There is emerging evidence that intermediate size droplets/particles (5-70um) can be transported considerable distances in indoor environments by air flows that might not be characterised as 'turbulent jets'. See for example https://doi.org/10.1063/5.0160579. I think this is important for the mechanistic viewpoints explored in this review. This is also relevant to Figure 1 which suggests that droplets >5um cannot travel beyond 1m. I think this is misleading and should be revised in the figure.

Figure 2 - The authors discuss droplet size and suggest residence times. While these trends are broadly accepted there are difficulties with suggesting that specific values apply to specific sizes. What are the environmental conditions under which these are assumed to apply? Are the sizes here the initial droplet size or the size after evaporation? What about droplets whose size is changing during their travel? How does the mixing of the air influence travel? How does the movement of occupants influence mixing and sedimentation (before resuspension of deposited material)? All of these factors make transport in indoor spaces more complex and result in a range of behaviours for a given initial droplet size. See Fig 6 in
https://doi.org/10.1111/ina.13146 for example.

p11, l174-177 - I think this sentence may be unhelpful and needs refining. It seems to imply that all droplets will settle under gravity. I think the difficulty stems from the binary definition used above (p10, l150-152) which seems to imply that droplets are above a certain size and below a certain size they are droplet nuclei. I believe it is more helpful to discuss respiratory emissions as droplets with a spectrum of sizes (see for example https://doi.org/10.1098/rsos.212022). Those that dry to sizes below 5um may be termed droplet nuclei but there is nothing distinct about the starting material. There is not a clear cut size below which droplets (dried or not) remain airborne and above which they deposit. Their airborne and other transport behaviour depends on the environmental conditions, including human activity.

p11, l182-l185 - The ability of larger sized droplets to reach the lungs is also not a binary function. Some larger droplets can reach beyond the head airways. They will of course carry much more virus than a smaller particle and this may be important for the probability of infection.

p11, l203 and discussion below this - It is important to challenge the identification of aerosols with those particles <5um. This is purely a definition of convenience. Particles suspended in the air can be much larger and travel beyond 1m as described above. Larger particles are difficult to measure.

p11, l210 - While it is true that smaller viral aerosols may be generated directly from infected individuals this is often difficult to separate in measurements from those that evaporate from droplets. Smaller particles are also likely to carry less virus, despite complex explanatory theories about different viral concentrations in the source fluid.

p12, l246-247 - And also conditions that favour intermediate size droplets / larger aerosols remaining airborne because of the human induced mixing (by movement or thermal effects).

p14, l318-329
up to p14, l316 - Another factor that is relevant in assessing the relative importance of fomites is the sampling efficiency from surfaces. This is rarely reported explicitly or used in the interpretation of surface samples but may be as low as a few percent.

Fig 3 - Very nice figure! It would be great if it showed human movement can contribute to resuspension but the text below addresses this directly. There is no need to change it.

p16-p17 - The authors make an argument for the importance of resuspension of particles in carrying virus. That seems reasonable. However, the section linking PM to infection seems unconvincing. I may have missed the point here but it seems to hint that PM may influence the mechanism of infection. I think that is also reasonable, but that would be expected to happen if PM (which is ubiquitous) is at high concentrations and doesn't rely on the PM actually carrying the virus (although it may also do so). The effect of PM on the respiratory cells doesn't require virus to be present on them to necessarily influence infection. I think this nuance is important to bring out to avoid the more important proposed mechanism (resuspension of virus bearing PM) being discounted.

p17, l469 - I'm not sure that there is a consensus that it is the upper respoiratory mucosa that are the most sensitive. Some have argued that the lower respiratory tract is important for serious illness.

p18, l504 - I think there has been a real polarisation in the debate about transmission mechanisms and an over-simplification in social media. I think it would be helpful to avoid reinforcing this polarisation by using different language here rather than talking of the 'droplets camp'.

Table 3 is not referenced in the text. It isn't clear what point it is making based on its position. This needs to be corrected.

Minor comments
p7, l30 - should be 'hygroscopic'

p11, l178 - word missing - 'by other'?

Table 1 - Despite their widespread use it isn't clear how Ct values relate to concentrations of RNA. Are you able to comment on this? (Also for Table 2)

p16, l427 - 'airborne particles' - particulate is an adjective albeit frequently misused as a noun. Also, l430 - exposed 'to the particles' or 'to the particulate matter'. And l444

---

## Round 0.2 · accepted · Accept

The authors have satisfactorily addressed most of all review comments and made the necessary changes to the manuscript.

Reviewer 2 ·

Basic reporting

Thanks to the authors for carefully addressed the comments from the first set of reviews (and apologies for missing the reference to Table 3 first time :) ). The changes have brought out some nuances that were missing before and I think it makes for a very rounded and important review. I believe the manuscript is ready for publication.

I spotted some minor, mostly typographical issues that I think can be addressed in the final edits.

Title & l95, l463 & l578 - 'aerial particles' or 'aerial particulate matter'

l28 - caps for 'Relative humidity'
l33 & l561 - indoors plural?
l278 - space needed between RH & and
l368 - missing full stop
l459 - 'reduces' plural

Experimental design

No comment

Validity of the findings

No comment

Additional comments

No comment